# The Generalized Reparameterization Gradient

**Francisco J. R. Ruiz**
University of Cambridge
Columbia University

**Michalis K. Titsias**
Athens University of
Economics and Business

**David M. Blei**
Columbia University

## Abstract

The reparameterization gradient has become a widely used method to obtain Monte Carlo gradients to optimize the variational objective. However, this technique does not easily apply to commonly used distributions such as beta or gamma without further approximations, and most practical applications of the reparameterization gradient fit Gaussian distributions. In this paper, we introduce *the generalized reparameterization gradient*, a method that extends the reparameterization gradient to a wider class of variational distributions. Generalized reparameterizations use invertible transformations of the latent variables which lead to transformed distributions that weakly depend on the variational parameters. This results in new Monte Carlo gradients that combine reparameterization gradients and score function gradients. We demonstrate our approach on variational inference for two complex probabilistic models. The generalized reparameterization is effective: even a single sample from the variational distribution is enough to obtain a low-variance gradient.

## 1 Introduction

Variational inference (VI) is a technique for approximating the posterior distribution in probabilistic models (Jordan et al., 1999; Wainwright and Jordan, 2008). Given a probabilistic model $p(\mathbf{x}, \mathbf{z})$ of observed variables $\mathbf{x}$ and hidden variables $\mathbf{z}$, the goal of VI is to approximate the posterior $p(\mathbf{z} \mid \mathbf{x})$, which is intractable to compute exactly for many models. The idea of VI is to posit a family of distributions over the latent variables $q(\mathbf{z}; \mathbf{v})$ with free variational parameters $\mathbf{v}$. VI then fits those parameters to find the member of the family that is closest in Kullback-Leibler (KL) divergence to the exact posterior, $\mathbf{v}^* = \arg\min_{\mathbf{v}} \mathrm{KL}(q(\mathbf{z}; \mathbf{v}) \| p(\mathbf{z} \mid \mathbf{x}))$. This turns inference into optimization, and different ways of doing VI amount to different optimization algorithms for solving this problem.

For a certain class of probabilistic models, those where each conditional distribution is in an exponential family, we can easily use coordinate ascent optimization to minimize the KL divergence (Ghahramani and Beal, 2001). However, many important models do not fall into this class (e.g., probabilistic neural networks or Bayesian generalized linear models). This is the scenario that we focus on in this paper. Much recent research in VI has focused on these difficult settings, seeking effective optimization algorithms that can be used with any model. This has enabled the application of VI on nonconjugate probabilistic models (Carbonetto et al., 2009; Paisley et al., 2012; Ranganath et al., 2014; Titsias and Lázaro-Gredilla, 2014), deep neural networks (Neal, 1992; Hinton et al., 1995; Mnih and Gregor, 2014; Kingma and Welling, 2014), and probabilistic programming (Wingate and Weber, 2013; Kucukelbir et al., 2015; van de Meent et al., 2016).

One strategy for VI in nonconjugate models is to obtain Monte Carlo estimates of the gradient of the variational objective and to use stochastic optimization to fit the variational parameters. Within this strategy, there have been two main lines of research: black-box variational inference (BBVI) (Ranganath et al., 2014) and reparameterization gradients (Salimans and Knowles, 2013; Kingma and Welling, 2014). Each enjoys different advantages and limitations.

BBVI expresses the gradient of the variational objective as an expectation with respect to the variational distribution using the log-derivative trick, also called REINFORCE or score function method (Glynn, 1990; Williams, 1992). It then takes samples from the variational distribution to calculate noisy gradients. BBVI is generic—it can be used with any type of latent variables and any model. However,

the gradient estimates typically suffer from high variance, which can lead to slow convergence. Ranganath et al. (2014) reduce the variance of these estimates using Rao-Blackwellization (Casella and Robert, 1996) and control variates (Ross, 2002; Paisley et al., 2012; Gu et al., 2016). Other researchers have proposed further reductions, e.g., through local expectations (Titsias and Lázaro-Gredilla, 2015) and importance sampling (Ruiz et al., 2016).

The second approach to Monte Carlo gradients of the variational objective is through reparameterization (Price, 1958; Bonnet, 1964; Salimans and Knowles, 2013; Kingma and Welling, 2014; Rezende et al., 2014). This approach reparameterizes the latent variable $\mathbf{z}$ in terms of a set of auxiliary random variables whose distributions do not depend on the variational parameters (typically, a standard normal). This facilitates taking gradients of the variational objective because the gradient operator can be pushed inside the expectation, and because the resulting procedure only requires drawing samples from simple distributions, such as standard normals. We describe this in detail in Section 2.

Reparameterization gradients exhibit lower variance than BBVI gradients. They typically need only one Monte Carlo sample to estimate a noisy gradient, which leads to fast algorithms. Further, for some models, their variance can be bounded (Fan et al., 2015). However, reparameterization is not as generic as BBVI. It is typically used with Gaussian variational distributions and does not easily generalize to other common distributions, such as the gamma or beta, without using further approximations. (See Knowles (2015) for an alternative approach to deal with the gamma distribution.)

We develop *the generalized reparameterization (G-REP) gradient*, a new method to extend reparameterization to other variational distributions. The main idea is to define an invertible transformation of the latent variables such that the distribution of the transformed variables is only weakly governed by the variational parameters. (We make this precise in Section 3.) Our technique naturally combines both BBVI and reparameterization; it applies to a wide class of nonconjugate models; it maintains the black-box criteria of reusing variational families; and it avoids approximations. We empirically show in two probabilistic models—a nonconjugate factorization model and a deep exponential family (Ranganath et al., 2015)—that a single Monte Carlo sample is enough to build an effective low-variance estimate of the gradient. In terms of speed, G-REP outperforms BBVI. In terms of accuracy, it outperforms automatic differentiation variational inference (ADVI) (Kucukelbir et al., 2016), which considers Gaussian variational distributions on a transformed space.

## 2 Background

Consider a probabilistic model $p(\mathbf{x}, \mathbf{z})$, where $\mathbf{z}$ denotes the latent variables and $\mathbf{x}$ the observations. We assume that the posterior distribution $p(\mathbf{z} \mid \mathbf{x})$ is analytically intractable and we wish to apply VI. We introduce a tractable distribution $q(\mathbf{z}; \mathbf{v})$ to approximate $p(\mathbf{z} \mid \mathbf{x})$ and minimize the KL divergence $D_{\mathrm{KL}}(q(\mathbf{z}; \mathbf{v}) \parallel p(\mathbf{z} \mid \mathbf{x}))$ with respect to the variational parameters $\mathbf{v}$. This minimization is equivalently expressed as the maximization of the so-called evidence lower bound (ELBO) (Jordan et al., 1999),

$$\mathcal{L}(\mathbf{v}) = \mathbb{E}_{q(\mathbf{z};\mathbf{v})}\left[\log p(\mathbf{x}, \mathbf{z}) - \log q(\mathbf{z}; \mathbf{v})\right] = \mathbb{E}_{q(\mathbf{z};\mathbf{v})}\left[f(\mathbf{z})\right] + \mathbb{H}\left[q(\mathbf{z}; \mathbf{v})\right]. \tag{1}$$

We denote

$$f(\mathbf{z}) \triangleq \log p(\mathbf{x}, \mathbf{z}) \tag{2}$$

to be the model log-joint density and $\mathbb{H}\left[q(\mathbf{z}; \mathbf{v})\right]$ to be the entropy of the variational distribution. When the expectation $\mathbb{E}_{q(\mathbf{z};\mathbf{v})}\left[f(\mathbf{z})\right]$ is analytically tractable, the maximization of the ELBO can be carried out using standard optimization methods. Otherwise, when it is intractable, other techniques are needed. Recent approaches rely on stochastic optimization to construct Monte Carlo estimates of the gradient with respect to the variational parameters. Below, we review the two main methods for building such Monte Carlo estimates: the score function method and the reparameterization trick.

**Score function method.** A general way to obtain unbiased stochastic gradients is to use the score function method, also called log-derivative trick or REINFORCE (Williams, 1992; Glynn, 1990), which has been recently applied to VI (Paisley et al., 2012; Ranganath et al., 2014; Mnih and Gregor, 2014). It is based on writing the gradient of the ELBO with respect to $\mathbf{v}$ as

$$\nabla_{\mathbf{v}}\mathcal{L} = \mathbb{E}_{q(\mathbf{z};\mathbf{v})}\left[f(\mathbf{z})\nabla_{\mathbf{v}}\log q(\mathbf{z}; \mathbf{v}) + \nabla_{\mathbf{v}}\mathbb{H}\left[q(\mathbf{z}; \mathbf{v})\right]\right], \tag{3}$$

and then building Monte Carlo estimates by approximating the expectation with samples from $q(\mathbf{z}; \mathbf{v})$. The resulting estimator suffers from high variance, making it necessary to apply variance reduction methods such as control variates (Ross, 2002) or Rao-Blackwellization (Casella and Robert, 1996). Such variance reduction techniques have been used in BBVI (Ranganath et al., 2014).

**Reparameterization.** The reparameterization trick (Salimans and Knowles, 2013; Kingma and Welling, 2014) expresses the latent variables $\mathbf{z}$ as an invertible function of another set of variables $\boldsymbol{\epsilon}$, i.e., $\mathbf{z} = \mathcal{T}(\boldsymbol{\epsilon}; \mathbf{v})$, such that the distribution of the new random variables $q_{\boldsymbol{\epsilon}}(\boldsymbol{\epsilon})$ does not depend on the variational parameters $\mathbf{v}$. Under these assumptions, expectations with respect to $q(\mathbf{z}; \mathbf{v})$ can be expressed as $\mathbb{E}_{q(\mathbf{z};\mathbf{v})}[f(\mathbf{z})] = \mathbb{E}_{q_{\boldsymbol{\epsilon}}(\boldsymbol{\epsilon})}[f(\mathcal{T}(\boldsymbol{\epsilon}; \mathbf{v}))]$, and the gradient with respect to $\mathbf{v}$ can be pushed into the expectation, yielding

$$\nabla_{\mathbf{v}}\mathcal{L} = \mathbb{E}_{q_{\boldsymbol{\epsilon}}(\boldsymbol{\epsilon})}\left[\nabla_{\mathbf{z}}f(\mathbf{z})\big|_{\mathbf{z}=\mathcal{T}(\boldsymbol{\epsilon};\mathbf{v})}\nabla_{\mathbf{v}}\mathcal{T}(\boldsymbol{\epsilon};\mathbf{v})\right] + \nabla_{\mathbf{v}}\mathbb{H}\left[q(\mathbf{z};\mathbf{v})\right]. \tag{4}$$

The assumption here is that the log-joint $f(\mathbf{z})$ is differentiable. The gradient $\nabla_{\mathbf{z}}f(\mathbf{z})$ depends on the model, but it can be computed using automatic differentiation tools (Baydin et al., 2015). Monte Carlo estimates of the reparameterization gradient typically present much lower variance than those based on Eq. 3. In practice, a single sample from $q_{\boldsymbol{\epsilon}}(\boldsymbol{\epsilon})$ is enough to obtain a low-variance estimate.[1]

The reparameterization trick is thus a powerful technique to reduce the variance of the estimator, but it requires a transformation $\boldsymbol{\epsilon} = \mathcal{T}^{-1}(\mathbf{z}; \mathbf{v})$ such that $q_{\boldsymbol{\epsilon}}(\boldsymbol{\epsilon})$ does not depend on the variational parameters $\mathbf{v}$. For instance, if the variational distribution is Gaussian with mean $\boldsymbol{\mu}$ and covariance $\boldsymbol{\Sigma}$, a straightforward transformation consists of standardizing the random variable $\mathbf{z}$, i.e.,

$$\boldsymbol{\epsilon} = \mathcal{T}^{-1}(\mathbf{z}; \boldsymbol{\mu}, \boldsymbol{\Sigma}) = \boldsymbol{\Sigma}^{-\frac{1}{2}}(\mathbf{z} - \boldsymbol{\mu}). \tag{5}$$

This transformation ensures that the (Gaussian) distribution $q_{\boldsymbol{\epsilon}}(\boldsymbol{\epsilon})$ does not depend on $\boldsymbol{\mu}$ or $\boldsymbol{\Sigma}$. For a general variational distribution $q(\mathbf{z}; \mathbf{v})$, Kingma and Welling (2014) discuss three families of transformations: inverse cumulative density function (CDF), location-scale, and composition. However, these transformations may not apply in certain cases.[2] Notably, none of them apply to the gamma[3] and the beta distributions, although these distributions are often used in VI.

Next, we show how to relax the constraint that the transformed density $q_{\boldsymbol{\epsilon}}(\boldsymbol{\epsilon})$ must not depend on the variational parameters $\mathbf{v}$. We follow a standardization procedure similar to the Gaussian case in Eq. 5, but we allow the distribution of the standardized variable $\boldsymbol{\epsilon}$ to depend (at least weakly) on $\mathbf{v}$.

# 3 The Generalized Reparameterization Gradient

We now generalize the reparameterization idea to distributions that, like the gamma or the beta, do not admit the standard reparameterization trick. We assume that we can efficiently sample from the variational distribution $q(\mathbf{z}; \mathbf{v})$, and that $q(\mathbf{z}; \mathbf{v})$ is differentiable with respect to $\mathbf{z}$ and $\mathbf{v}$. We introduce a random variable $\boldsymbol{\epsilon}$ defined by an invertible transformation

$$\boldsymbol{\epsilon} = \mathcal{T}^{-1}(\mathbf{z}; \mathbf{v}), \qquad \text{and} \qquad \mathbf{z} = \mathcal{T}(\boldsymbol{\epsilon}; \mathbf{v}), \tag{6}$$

where we can think of $\boldsymbol{\epsilon} = \mathcal{T}^{-1}(\mathbf{z}; \mathbf{v})$ as a *standardization procedure* that attempts to make the distribution of $\boldsymbol{\epsilon}$ weakly dependent on the variational parameters $\mathbf{v}$. "Weakly" means that at least its first moment does not depend on $\mathbf{v}$. For instance, if $\boldsymbol{\epsilon}$ is defined to have zero mean, then its first moment has become independent of $\mathbf{v}$. However, we *do not* assume that the resulting distribution of $\boldsymbol{\epsilon}$ is completely independent of the variational parameters $\mathbf{v}$, and therefore we write it as $q_{\boldsymbol{\epsilon}}(\boldsymbol{\epsilon}; \mathbf{v})$. We use the distribution $q_{\boldsymbol{\epsilon}}(\boldsymbol{\epsilon}; \mathbf{v})$ in the derivation of G-REP, but we write the final gradient as an expectation with respect to the original variational distribution $q(\mathbf{z}; \mathbf{v})$, from which we can sample.

More in detail, by the standard change-of-variable technique, the transformed density is

$$q_{\boldsymbol{\epsilon}}(\boldsymbol{\epsilon}; \mathbf{v}) = q(\mathcal{T}(\boldsymbol{\epsilon}; \mathbf{v}); \mathbf{v})\, J(\boldsymbol{\epsilon}, \mathbf{v}), \quad \text{where} \quad J(\boldsymbol{\epsilon}, \mathbf{v}) \triangleq |\det \nabla_{\boldsymbol{\epsilon}} \mathcal{T}(\boldsymbol{\epsilon}; \mathbf{v})|, \tag{7}$$

is a short-hand for the absolute value of the determinant of the Jacobian. We first use the transformation to rewrite the gradient of $\mathbb{E}_{q(\mathbf{z};\mathbf{v})}[f(\mathbf{z})]$ in (1) as

$$\nabla_{\mathbf{v}}\mathbb{E}_{q(\mathbf{z};\mathbf{v})}[f(\mathbf{z})] = \nabla_{\mathbf{v}}\mathbb{E}_{q_{\boldsymbol{\epsilon}}(\boldsymbol{\epsilon};\mathbf{v})}[f(\mathcal{T}(\boldsymbol{\epsilon};\mathbf{v}))] = \nabla_{\mathbf{v}}\int q_{\boldsymbol{\epsilon}}(\boldsymbol{\epsilon}; \mathbf{v})f(\mathcal{T}(\boldsymbol{\epsilon};\mathbf{v}))\,d\boldsymbol{\epsilon}. \tag{8}$$

We now express the gradient as the sum of two terms, which we name $\mathbf{g}^{\text{rep}}$ and $\mathbf{g}^{\text{corr}}$ for reasons that we will explain below. We apply the log-derivative trick and the product rule for derivatives, yielding

$$\nabla_{\mathbf{v}} \mathbb{E}_{q(\mathbf{z};\mathbf{v})}\left[f(\mathbf{z})\right] = \underbrace{\int q_{\boldsymbol{\epsilon}}(\boldsymbol{\epsilon};\mathbf{v})\nabla_{\mathbf{v}} f\left(\mathcal{T}(\boldsymbol{\epsilon};\mathbf{v})\right)d\boldsymbol{\epsilon}}_{\mathbf{g}^{\text{rep}}} + \underbrace{\int q_{\boldsymbol{\epsilon}}(\boldsymbol{\epsilon};\mathbf{v})f\left(\mathcal{T}(\boldsymbol{\epsilon};\mathbf{v})\right)\nabla_{\mathbf{v}}\log q_{\boldsymbol{\epsilon}}(\boldsymbol{\epsilon};\mathbf{v})d\boldsymbol{\epsilon}}_{\mathbf{g}^{\text{corr}}}, \quad (9)$$

We rewrite Eq. 9 as an expression that involves expectations with respect to the original variational distribution $q(\mathbf{z};\mathbf{v})$ only. For that, we define the following two auxiliary functions that depend on the transformation $\mathcal{T}(\boldsymbol{\epsilon};\mathbf{v})$:

$$h(\boldsymbol{\epsilon};\mathbf{v}) \triangleq \nabla_{\mathbf{v}}\mathcal{T}(\boldsymbol{\epsilon};\mathbf{v}), \qquad \text{and} \qquad u(\boldsymbol{\epsilon};\mathbf{v}) \triangleq \nabla_{\mathbf{v}}\log J(\boldsymbol{\epsilon},\mathbf{v}). \quad (10)$$

After some algebra (see the Supplement for details), we obtain

$$\mathbf{g}^{\text{rep}} = \mathbb{E}_{q(\mathbf{z};\mathbf{v})}\left[\nabla_{\mathbf{z}} f(\mathbf{z})h\left(\mathcal{T}^{-1}(\mathbf{z};\mathbf{v});\mathbf{v}\right)\right],$$
$$\mathbf{g}^{\text{corr}} = \mathbb{E}_{q(\mathbf{z};\mathbf{v})}\left[f(\mathbf{z})\left(\nabla_{\mathbf{z}}\log q(\mathbf{z};\mathbf{v})h\left(\mathcal{T}^{-1}(\mathbf{z};\mathbf{v});\mathbf{v}\right) + \nabla_{\mathbf{v}}\log q(\mathbf{z};\mathbf{v}) + u\left(\mathcal{T}^{-1}(\mathbf{z};\mathbf{v});\mathbf{v}\right)\right)\right]. \quad (11)$$

Thus, we can finally write the full gradient of the ELBO as

$$\nabla_{\mathbf{v}}\mathcal{L} = \mathbf{g}^{\text{rep}} + \mathbf{g}^{\text{corr}} + \nabla_{\mathbf{v}}\mathbb{H}\left[q(\mathbf{z};\mathbf{v})\right], \quad (12)$$

**Interpretation of the generalized reparameterization gradient.** The term $\mathbf{g}^{\text{rep}}$ is easily recognizable as the standard reparameterization gradient, and hence the label "rep." Indeed, if the distribution $q_{\boldsymbol{\epsilon}}(\boldsymbol{\epsilon};\mathbf{v})$ does not depend on the variational parameters $\mathbf{v}$, then the term $\nabla_{\mathbf{v}}\log q_{\boldsymbol{\epsilon}}(\boldsymbol{\epsilon};\mathbf{v})$ in Eq. 9 vanishes, making $\mathbf{g}^{\text{corr}} = \mathbf{0}$. Thus, we may interpret $\mathbf{g}^{\text{corr}}$ as a "correction" term that appears when the transformed density depends on the variational parameters.

Furthermore, we can recover the score function gradient in Eq. 3 by choosing the identity transformation, $\mathbf{z} = \mathcal{T}(\boldsymbol{\epsilon};\mathbf{v}) = \boldsymbol{\epsilon}$. In such case, the auxiliary functions in Eq. 10 become zero because the transformation does not depend on $\mathbf{v}$, i.e., $h(\boldsymbol{\epsilon};\mathbf{v}) = \mathbf{0}$ and $u(\boldsymbol{\epsilon};\mathbf{v}) = \mathbf{0}$. This implies that $\mathbf{g}^{\text{rep}} = \mathbf{0}$ and $\mathbf{g}^{\text{corr}} = \mathbb{E}_{q(\mathbf{z};\mathbf{v})}\left[f(\mathbf{z})\nabla_{\mathbf{v}}\log q(\mathbf{z};\mathbf{v})\right]$.

Alternatively, we can interpret the G-REP gradient as a control variate of the score function gradient. For that, we rearrange Eqs. 9 and 11 to express the gradient as

$$\nabla_{\mathbf{v}}\mathbb{E}_{q(\mathbf{z};\mathbf{v})}\left[f(\mathbf{z})\right] = \mathbb{E}_{q(\mathbf{z};\mathbf{v})}\left[f(\mathbf{z})\nabla_{\mathbf{v}}\log q(\mathbf{z};\mathbf{v})\right]$$
$$+ \mathbf{g}^{\text{rep}} + \mathbb{E}_{q(\mathbf{z};\mathbf{v})}\left[f(\mathbf{z})\left(\nabla_{\mathbf{z}}\log q(\mathbf{z};\mathbf{v})h\left(\mathcal{T}^{-1}(\mathbf{z};\mathbf{v});\mathbf{v}\right) + u\left(\mathcal{T}^{-1}(\mathbf{z};\mathbf{v});\mathbf{v}\right)\right)\right],$$

where the second line is the control variate, which involves the reparameterization gradient.

**Transformations.** Eqs. 9 and 11 are valid for any transformation $\mathcal{T}(\boldsymbol{\epsilon};\mathbf{v})$. However, we may expect some transformations to perform better than others, in terms of the variance of the resulting estimator. It seems sensible to search for transformations that make $\mathbf{g}^{\text{corr}}$ small, as the reparameterization gradient $\mathbf{g}^{\text{rep}}$ is known to present low variance in practice under standard smoothness conditions of the log-joint (Fan et al., 2015).[4] Transformations that make $\mathbf{g}^{\text{corr}}$ small are such that $\boldsymbol{\epsilon} = \mathcal{T}^{-1}(\mathbf{z};\mathbf{v})$ becomes weakly dependent on the variational parameters $\mathbf{v}$. In the standard reparameterization of Gaussian random variables, the transformation takes the form in (5), and thus $\boldsymbol{\epsilon}$ is a standardized version of $\mathbf{z}$. We mimic this standardization idea for other distributions as well. In particular, for exponential family distributions, we use transformations of the form (sufficient statistic − expected sufficient statistic)/(scale factor). We present several examples in the next section.

### 3.1 Examples

For concreteness, we show here some examples of the equations above for well-known probability distributions. In particular, we choose the gamma, log-normal, and beta distributions.

**Gamma distribution.** Let $q(z;\alpha,\beta)$ be a gamma distribution with shape $\alpha$ and rate $\beta$. We use a transformation based on standardization of the sufficient statistic $\log(z)$, i.e.,

$$\epsilon = \mathcal{T}^{-1}(z;\alpha,\beta) = \frac{\log(z) - \psi(\alpha) + \log(\beta)}{\sqrt{\psi_1(\alpha)}},$$

where $\psi(\cdot)$ denotes the digamma function, and $\psi_k(\cdot)$ is its $k$-th derivative. This ensures that $\epsilon$ has zero mean and unit variance, and thus its two first moments do not depend on the variational parameters $\alpha$ and $\beta$. We now compute the auxiliary functions in Eq. 10 for the components of the gradient with respect to $\alpha$ and $\beta$, which take the form

$$h_\alpha(\epsilon; \alpha, \beta) = \mathcal{T}(\epsilon; \alpha, \beta) \left( \frac{\epsilon \psi_2(\alpha)}{2\sqrt{\psi_1(\alpha)}} + \psi_1(\alpha) \right), \qquad h_\beta(\epsilon; \alpha, \beta) = -\frac{\mathcal{T}(\epsilon; \alpha, \beta)}{\beta},$$

$$u_\alpha(\epsilon; \alpha, \beta) = \left( \frac{\epsilon \psi_2(\alpha)}{2\sqrt{\psi_1(\alpha)}} + \psi_1(\alpha) \right) + \frac{\psi_2(\alpha)}{2\psi_1(\alpha)}, \qquad u_\beta(\epsilon; \alpha, \beta) = -\frac{1}{\beta}.$$

The terms $\mathbf{g}^{\mathrm{rep}}$ and $\mathbf{g}^{\mathrm{corr}}$ are obtained after substituting these results in Eq. 11. We provide the final expressions in the Supplement. We remark here that the component of $\mathbf{g}^{\mathrm{corr}}$ corresponding to the derivative with respect to the rate equals zero, i.e., $\mathbf{g}_\beta^{\mathrm{corr}} = 0$, meaning that the distribution of $\epsilon$ does not depend on the parameter $\beta$. Indeed, we can compute this distribution following Eq. 7 as

$$q_\epsilon(\epsilon; \alpha, \beta) = \frac{e^{\alpha\psi(\alpha)}\sqrt{\psi_1(\alpha)}}{\Gamma(\alpha)} \exp\left( \epsilon\alpha\sqrt{\psi_1(\alpha)} - \exp\left( \epsilon\sqrt{\psi_1(\alpha)} + \psi(\alpha) \right) \right),$$

where we can verify that it does not depend on $\beta$.

**Log-normal distribution.** For a log-normal distribution with location $\mu$ and scale $\sigma$, we can standardize the sufficient statistic $\log(z)$ as

$$\epsilon = \mathcal{T}^{-1}(z; \mu, \sigma) = \frac{\log(z) - \mu}{\sigma}.$$

This leads to a standard normal distribution on $\epsilon$, which does not depend on the variational parameters, and thus $\mathbf{g}^{\mathrm{corr}} = \mathbf{0}$. The auxiliary function $h(\epsilon; \mu, \sigma)$, which is needed for $\mathbf{g}^{\mathrm{rep}}$, takes the form

$$h_\mu(\epsilon; \mu, \sigma) = \mathcal{T}(\epsilon; \mu, \sigma), \qquad h_\sigma(\epsilon; \mu, \sigma) = \epsilon\mathcal{T}(\epsilon; \mu, \sigma).$$

Thus, the reparameterization gradient is given in this case by

$$\mathbf{g}_\mu^{\mathrm{rep}} = \mathbb{E}_{q(z;\mu,\sigma)}\left[ z\nabla_z f(\mathbf{z}) \right], \qquad \mathbf{g}_\sigma^{\mathrm{rep}} = \mathbb{E}_{q(z;\mu,\sigma)}\left[ z\mathcal{T}^{-1}(z; \mu, \sigma)\nabla_z f(\mathbf{z}) \right].$$

This corresponds to ADVI (Kucukelbir et al., 2016) with a logarithmic transformation over a positive random variable, since the variational distribution over the transformed variable is Gaussian. For a general variational distribution, we recover ADVI if the transformation makes $\epsilon$ Gaussian.

**Beta distribution.** For a random variable $z \sim \mathrm{Beta}(\alpha, \beta)$, we could rewrite $z = z_1'/(z_1' + z_2')$ for $z_1' \sim \mathrm{Gamma}(\alpha, 1)$ and $z_2' \sim \mathrm{Gamma}(\beta, 1)$, and apply the gamma reparameterization for $z_1'$ and $z_2'$. Instead, in the spirit of applying standardization directly over $z$, we define a transformation to standardize the logit function, $\mathrm{logit}(z) \triangleq \log(z/(1-z))$ (sum of sufficient statistics of the beta),

$$\epsilon = \mathcal{T}^{-1}(z; \alpha, \beta) = \frac{\mathrm{logit}(z) - \psi(\alpha) + \psi(\beta)}{\sigma(\alpha, \beta)}.$$

This ensures that $\epsilon$ has zero mean. We can set the denominator to the standard deviation of $\mathrm{logit}(z)$. However, for larger-scaled models we found better performance with a denominator $\sigma(\alpha, \beta)$ that makes $\mathbf{g}^{\mathrm{corr}} = \mathbf{0}$ for the currently drawn sample $z$ (see the Supplement for details), even though the variance of the transformed variable $\epsilon$ is not one in such case.[5] The reason is that $\mathbf{g}^{\mathrm{corr}}$ suffers from high variance in the same way as the score function estimator does.

## 3.2 Algorithm

We now present our full algorithm for G-REP. It requires the specification of the variational family and the transformation $\mathcal{T}(\epsilon; \mathbf{v})$. Given these, the full procedure is summarized in Algorithm 1. We use the adaptive step-size sequence proposed by Kucukelbir et al. (2016), which combines RMSPROP (Tieleman and Hinton, 2012) and Adagrad (Duchi et al., 2011). Let $g_k^{(i)}$ be the $k$-th component of the gradient at the $i$-th iteration, and $\rho_k^{(i)}$ the step-size for that component. We set

$$\rho_k^{(i)} = \eta \times i^{-0.5+\kappa} \times \left( \tau + \sqrt{s_k^{(i)}} \right)^{-1}, \qquad \text{with} \qquad s_k^{(i)} = \gamma(g_k^{(i)})^2 + (1-\gamma)s_k^{(i-1)}, \qquad (13)$$

where we set $\kappa = 10^{-16}$, $\tau = 1$, $\gamma = 0.1$, and we explore several values of $\eta$. Thus, we update the variational parameters as $\mathbf{v}^{(i+1)} = \mathbf{v}^{(i)} + \boldsymbol{\rho}^{(i)} \circ \nabla_{\mathbf{v}}\mathcal{L}$, where '$\circ$' is the element-wise product.

[5]Note that this introduces some bias since we are ignoring the dependence of $\sigma(\alpha, \beta)$ on $z$.

**Algorithm 1:** Generalized reparameterization gradient algorithm

---

**input** : data $\mathbf{x}$, probabilistic model $p(\mathbf{x}, \mathbf{z})$, variational family $q(\mathbf{z}; \mathbf{v})$, transformation $\mathbf{z} = \mathcal{T}(\boldsymbol{\epsilon}; \mathbf{v})$
**output** : variational parameters $\mathbf{v}$
Initialize $\mathbf{v}$
**repeat**
    Draw a single sample $\mathbf{z} \sim q(\mathbf{z}; \mathbf{v})$
    Compute the auxiliary functions $h\left(\mathcal{T}^{-1}(\mathbf{z}; \mathbf{v}); \mathbf{v}\right)$ and $u\left(\mathcal{T}^{-1}(\mathbf{z}; \mathbf{v}); \mathbf{v}\right)$ (Eq. 10)
    Estimate $\mathbf{g}^{\text{rep}}$ and $\mathbf{g}^{\text{corr}}$ (Eq. 11, estimate the expectation with one sample)
    Compute (analytic) or estimate (Monte Carlo) the gradient of the entropy, $\nabla_{\mathbf{v}} \mathbb{H}[q(\mathbf{z}; \mathbf{v})]$
    Compute the noisy gradient $\nabla_{\mathbf{v}} \mathcal{L}$ (Eq. 12)
    Set the step-size $\boldsymbol{\rho}^{(i)}$ (Eq. 13) and take a gradient step for $\mathbf{v}$
**until** *convergence*

---

### 3.3 Related work

A closely related VI method is ADVI, which also relies on reparameterization and has been incorporated into Stan (Kucukelbir et al., 2015, 2016). ADVI applies a transformation to the random variables such that their support is on the reals and then uses a Gaussian variational posterior on the transformed space. For instance, random variables that are constrained to be positive are first transformed through a logarithmic function and then a Gaussian variational approximating distribution is placed on the unconstrained space. Thus, ADVI struggles to approximate probability densities with singularities, which are useful in models where sparsity is appropriate. In contrast, the G-REP method allows to estimate the gradient for a wider class of variational distributions, including gamma and beta distributions, which are more appropriate to encode sparsity constraints.

Schulman et al. (2015) also write the gradient in the form given in Eq. 12 to automatically estimate the gradient through a backpropagation algorithm in the context of stochastic computation graphs. However, they do not provide additional insight into this equation, do not apply it to general VI, do not discuss transformations for any distributions, and do not report experiments. Thus, our paper complements Schulman et al. (2015) and provides an off-the-shelf tool for general VI.

## 4 Experiments

We apply G-REP to perform mean-field VI on two nonconjugate probabilistic models: the sparse gamma deep exponential family (DEF) and a beta-gamma matrix factorization (MF) model. The sparse gamma DEF (Ranganath et al., 2015) is a probabilistic model with several layers of latent locations and latent weights, mimicking the architecture of a deep neural network. The weights of the model are denoted by $w_{kk'}^{(\ell)}$, where $k$ and $k'$ run over latent components, and $\ell$ indexes the layer. The latent locations are $z_{nk}^{(\ell)}$, where $n$ denotes the observation. We consider Poisson-distributed observations $x_{nd}$ for each dimension $d$. Thus, the model is specified as

$$z_{nk}^{(\ell)} \sim \text{Gamma}\left(\alpha_z, \frac{\alpha_z}{\sum_{k'} z_{nk'}^{(\ell+1)} w_{k'k}^{(\ell)}}\right), \qquad x_{nd} \sim \text{Poisson}\left(\sum_{k'} z_{nk'}^{(1)} w_{k'd}^{(0)}\right).$$

We place gamma priors over the weights $w_{kk'}^{\ell}$ with rate 0.3 and shape 0.1, and a gamma prior with rate 0.1 and shape 0.1 over the top-layer latent variables $z_{nk}^{(L)}$. We set the hyperparameter $\alpha_z = 0.1$, and we use $L = 3$ layers with 100, 40, and 15 latent factors.

The second model is a beta-gamma MF model with weights $w_{kd}$ and latent locations $z_{nk}$. We use this model to describe binary observations $x_{nd}$, which are modeled as

$$x_{nd} \sim \text{Bernoulli}\left(\text{sigmoid}\left(\sum_k \text{logit}(z_{nk}) w_{kd}\right)\right),$$

where $\text{logit}(z) = \log(z/(1-z))$ and $\text{sigmoid}(\cdot)$ is the inverse logit function. We place a gamma prior with shape 0.1 and rate 0.3 over the weights $w_{kd}$, a uniform prior over the variables $z_{nk}$, and we use $K = 100$ latent components.

**Datasets.** We apply the sparse gamma DEF on two different databases: (i) the Olivetti database at AT&T,[6] which consists of 400 (320 for training and 80 for test) $64 \times 64$ images of human faces in a 8

| Dataset | G-REP | BBVI | ADVI | Dataset | G-REP | BBVI | ADVI |
|---------|-------|------|------|---------|-------|------|------|
| Olivetti | 5 | 1 | 0.1 | Olivetti | 0.46 | 12.90 | 0.17 |
| NIPS | 0.5 | 5 | 1 | NIPS | 0.83 | 20.95 | 0.25 |
| MNIST | 5 | 5 | 0.1 | MNIST | 1.09 | 25.99 | 0.34 |
| Omniglot | 5 | – | 0.1 | Omniglot | 5.50 | – | 4.10 |

**Table 1:** (Left) Step-size constant $\eta$, reported for completeness. (Right) Average time per iteration in seconds. G-REP is 1-4 times slower than ADVI but above one order of magnitude faster than BBVI.

bit scale $(0 - 255)$; and (ii) the collection of papers at the Neural Information Processing Systems (NIPS) 2011 conference, which consists of 305 documents and a vocabulary of 5715 effective words in a bag-of-words format (25% of words from all documents are set aside to form the test set).

We apply the beta-gamma MF on: (i) the binarized MNIST data,[7] which consists of $28 \times 28$ images of hand-written digits (we use 5000 training and 2000 test images); and (ii) the Omniglot dataset (Lake et al., 2015), which consists of $105 \times 105$ images of hand-written characters from different alphabets (we select 10 alphabets, with 4425 training images, 1475 test images, and 295 characters).

**Evaluation.** We apply mean-field VI and we compare G-REP with BBVI (Ranganath et al., 2014) and ADVI (Kucukelbir et al., 2016). We do not apply BBVI on the Omniglot dataset due to its computational complexity. At each iteration, we evaluate the ELBO using one sample from the variational distribution, except for ADVI, for which we use 20 samples (for the Omniglot dataset, we only use one sample). We run each algorithm with a fixed computational budget of CPU time. After that time, we also evaluate the predictive log-likelihood on the test set, averaging over 100 posterior samples. For the NIPS data, we also compute the test perplexity (with one posterior sample) every 10 iterations, given by

$$\exp\left( \frac{- \sum_{\text{docs}} \sum_{w \in \text{doc}(d)} \log p(w \,|\, \#\text{held out in doc}(d))}{\#\text{held out words}} \right).$$

**Experimental setup.** To estimate the gradient, we use 30 Monte Carlo samples for BBVI, and only 1 for ADVI and G-REP. For BBVI, we use Rao-Blackwellization and control variates (we use a separate set of 30 samples to estimate the control variates). For BBVI and G-REP, we use beta and gamma variational distributions, whereas ADVI uses Gaussian distributions on the transformed space, which correspond to log-normal or logit-normal distributions on the original space. Thus, only G-REP and BBVI optimize the same variational family. We parameterize the gamma distribution in terms of its shape and mean, and the beta in terms of its shape parameters $\alpha$ and $\beta$. To avoid constrained optimization, we apply the transformation $v' = \log(\exp(v) - 1)$ to the variational parameters that are constrained to be positive and take stochastic gradient steps with respect to $v'$. We use the analytic gradient of the entropy terms. We implement ADVI as described by Kucukelbir et al. (2016).

We use the step-size schedule in Eq. 13, and we explore the parameter $\eta \in \{0.1, 0.5, 1, 5\}$. For each algorithm and each dataset, we report the results based on the value of $\eta$ for which the best ELBO was achieved. We report the values of $\eta$ in Table 1 (left).

**Results.** We show in Figure 1 the evolution of the ELBO as a function of the running time for three of the considered datasets. BBVI converges slower than the rest of the methods, since each iteration involves drawing multiple samples and evaluating the log-joint for each of them. ADVI and G-REP achieve similar bounds, except for the MNIST dataset, for which G-REP provides a variational approximation that is closer to the posterior, since the ELBO is higher. This is because a variational family with sparse gamma and beta distributions provides a better fit to the data than the variational family to which ADVI is limited (log-normal and logit-normal). ADVI seems to converge slower; however, we *do not* claim that ADVI converges slower than G-REP in general. Instead, the difference may be due to the different step-sizes schedules that we found to be optimal (see Table 1). We also report in Table 1 (right) the average time per iteration[8] for each method: BBVI is the slowest method, and ADVI is the fastest because it involves simulation of Gaussian random variables only.

However, G-REP provides higher likelihood values than ADVI. We show in Figure 2a the evolution of the perplexity (lower is better) for the NIPS dataset, and in Figure 2b the resulting test log-likelihood (larger is better) for the rest of the considered datasets. In Figure 2b, we report the mean and standard deviation over 100 posterior samples. ADVI cannot fit the data as well as G-REP or BBVI because it is constrained to log-normal and logit-normal variational distributions. These cannot capture sparsity,

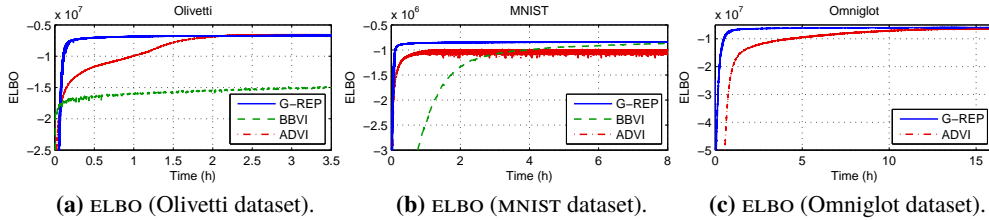

**(a)** ELBO (Olivetti dataset).      **(b)** ELBO (MNIST dataset).      **(c)** ELBO (Omniglot dataset).

**Figure 1:** Comparison between G-REP, BBVI, and ADVI in terms of the variational objective function.

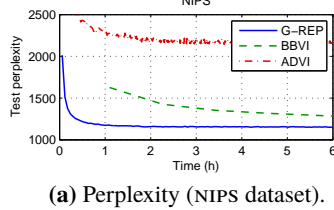

**(a)** Perplexity (NIPS dataset).

| Dataset | G-REP | BBVI | ADVI |
|---------|-------|------|------|
| Olivetti | $-\mathbf{4.48 \pm 0.01}$ | $-9.74 \pm 0.08$ | $-4.63 \pm 0.01$ |
| MNIST | $-0.0932 \pm 0.0004$ | $-\mathbf{0.0888 \pm 0.0004}$ | $-0.189 \pm 0.009$ |
| Omniglot | $-\mathbf{0.0472 \pm 0.0001}$ | $-$ | $-0.0823 \pm 0.0009$ |

**(b)** Average test log-likelihood per entry $x_{nd}$.

**Figure 2:** Comparison between G-REP, BBVI, and ADVI in terms of performance on the test set. G-REP outperforms BBVI because the latter has not converged in the allowed time, and it also outperforms ADVI because of the variational family it uses.

which is an important feature for the considered models. We can also conclude this by a simple visual inspection of the fitted models. In the Supplement, we compare images sampled from the G-REP and the ADVI posteriors, where we can observe that the latter are more blurry or lack some details.

## 5 Conclusion

We have introduced the generalized reparameterization gradient (G-REP), a technique to extend the standard reparameterization gradient to a wider class of variational distributions. As the standard reparameterization method, our method is applicable to any probabilistic model that is differentiable with respect to the latent variables. We have demonstrated the generalized reparameterization gradient on two nonconjugate probabilistic models to fit a variational approximation involving gamma and beta distributions. We have also empirically shown that a single Monte Carlo sample is enough to obtain a noisy estimate of the gradient, therefore leading to a fast inference procedure.

### Acknowledgments

This project has received funding from the EU H2020 programme (Marie Skłodowska-Curie grant agreement 706760), NFS IIS-1247664, ONR N00014-11-1-0651, DARPA FA8750-14-2-0009, DARPA N66001-15-C-4032, Adobe, the John Templeton Foundation, and the Sloan Foundation. The authors would also like to thank Kriste Krstovski, Alp Kuckukelbir, and Christian A. Naesseth for helpful comments and discussions.

## Footnotes

[1] In the literature, there is no formal proof that reparameterization has lower variance than the score function estimator, except for some simple models (Fan et al., 2015). Titsias and Lázaro-Gredilla (2014) provide some intuitions, and Rezende et al. (2014) show some benefits of reparameterization in the Gaussian case.

[2] The inverse CDF approach sets $\mathcal{T}^{-1}(\mathbf{z}; \mathbf{v})$ to the CDF. This leads to a uniform distribution over $\boldsymbol{\epsilon}$ on the unit interval, but it is not practical because the inverse CDF, $\mathcal{T}(\boldsymbol{\epsilon}; \mathbf{v})$, does not have analytical solution in general. We develop an approach that does not require computation of (inverse) CDF's or their derivatives.

[3] Composition is only available when it is possible to express the gamma as a sum of exponentials, i.e., its shape parameter is an integer, which is not generally the case in VI.

[4]Techniques such as Rao-Blackwellization could additionally be applied to reduce the variance of $\mathbf{g}^{\text{corr}}$. We do *not* apply any such technique in this paper.

[6] http://www.cl.cam.ac.uk/research/dtg/attarchive/facedatabase.html

[7]http://yann.lecun.com/exdb/mnist

[8]On the full MNIST with $50,000$ training images, G-REP (ADVI) took 8.08 (2.04) seconds per iteration.

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
