[Supplementary Material · supp_mat.pdf]

# The Generalized Reparameterization Gradient: Supplement

**Francisco J. R. Ruiz**
University of Cambridge
Columbia University

**Michalis K. Titsias**
Athens University of
Economics and Business

**David M. Blei**
Columbia University

## 1 Derivation of the Generalized Reparameterization Gradient

Here we show the mathematical derivation of the generalized reparameterization gradient. Firstly, recall the definition of the functions

$$h(\boldsymbol{\epsilon}; \mathbf{v}) \triangleq \nabla_{\mathbf{v}} \mathcal{T}(\boldsymbol{\epsilon}; \mathbf{v}), \tag{1}$$

$$u(\boldsymbol{\epsilon}; \mathbf{v}) \triangleq \nabla_{\mathbf{v}} \log J(\boldsymbol{\epsilon}, \mathbf{v}), \tag{2}$$

which are provided in the main text.

We start from the following expression of the gradient, also derived in the main text:

$$\nabla_{\mathbf{v}} \mathbb{E}_{q(\mathbf{z};\mathbf{v})} \left[ f(\mathbf{z}) \right] = \underbrace{\int q_{\boldsymbol{\epsilon}}(\boldsymbol{\epsilon}; \mathbf{v}) \nabla_{\mathbf{v}} f\left(\mathcal{T}(\boldsymbol{\epsilon}; \mathbf{v})\right) d\boldsymbol{\epsilon}}_{\mathbf{g}^{\text{rep}}} + \underbrace{\int q_{\boldsymbol{\epsilon}}(\boldsymbol{\epsilon}; \mathbf{v}) f\left(\mathcal{T}(\boldsymbol{\epsilon}; \mathbf{v})\right) \nabla_{\mathbf{v}} \log q_{\boldsymbol{\epsilon}}(\boldsymbol{\epsilon}; \mathbf{v}) d\boldsymbol{\epsilon}}_{\mathbf{g}^{\text{corr}}}, \tag{3}$$

We can write the former term, $\mathbf{g}^{\text{rep}}$, as

$$\mathbf{g}^{\text{rep}} = \int q_{\boldsymbol{\epsilon}}(\boldsymbol{\epsilon}; \mathbf{v}) \nabla_{\mathbf{v}} f\left(\mathcal{T}(\boldsymbol{\epsilon}; \mathbf{v})\right) d\boldsymbol{\epsilon} \tag{4}$$

$$= \int q\left(\mathcal{T}(\boldsymbol{\epsilon}; \mathbf{v}); \mathbf{v}\right) J(\boldsymbol{\epsilon}, \mathbf{v}) \nabla_{\mathbf{v}} f\left(\mathcal{T}(\boldsymbol{\epsilon}; \mathbf{v})\right) d\boldsymbol{\epsilon} \tag{5}$$

$$= \int q\left(\mathcal{T}(\boldsymbol{\epsilon}; \mathbf{v}); \mathbf{v}\right) J(\boldsymbol{\epsilon}, \mathbf{v}) \nabla_{\mathbf{z}} f(\mathbf{z})\big|_{\mathbf{z}=\mathcal{T}(\boldsymbol{\epsilon};\mathbf{v})} \nabla_{\mathbf{v}} \mathcal{T}(\boldsymbol{\epsilon}; \mathbf{v}) d\boldsymbol{\epsilon} \tag{6}$$

$$= \int q(\mathbf{z}; \mathbf{v}) \nabla_{\mathbf{z}} f(\mathbf{z}) h\left(\mathcal{T}^{-1}(\mathbf{z}; \mathbf{v}); \mathbf{v}\right) d\mathbf{z} \tag{7}$$

$$= \mathbb{E}_{q(\mathbf{z};\mathbf{v})} \left[ \nabla_{\mathbf{z}} f(\mathbf{z}) h\left(\mathcal{T}^{-1}(\mathbf{z}; \mathbf{v}); \mathbf{v}\right) \right], \tag{8}$$

where we have first replaced the variational distribution on the transformed space with its form as a function of $q(\mathbf{z}; \mathbf{v})$, i.e., $q_{\boldsymbol{\epsilon}}(\boldsymbol{\epsilon}; \mathbf{v}) = q\left(\mathcal{T}(\boldsymbol{\epsilon}; \mathbf{v}); \mathbf{v}\right) J(\boldsymbol{\epsilon}, \mathbf{v})$. We have then applied the chain rule, and finally we have made a new change of variables back to the original space $\mathbf{z}$ (thus multiplying by the inverse Jacobian).

For the latter, $\mathbf{g}^{\text{corr}}$, we have that

$$\mathbf{g}^{\text{corr}} = \int q_{\boldsymbol{\epsilon}}(\boldsymbol{\epsilon}; \mathbf{v}) f\left(\mathcal{T}(\boldsymbol{\epsilon}; \mathbf{v})\right) \nabla_{\mathbf{v}} \log q_{\boldsymbol{\epsilon}}(\boldsymbol{\epsilon}; \mathbf{v}) d\boldsymbol{\epsilon} \tag{9}$$

$$= \int q\left(\mathcal{T}(\boldsymbol{\epsilon}, \mathbf{v}); \mathbf{v}\right) J(\boldsymbol{\epsilon}, \mathbf{v}) f\left(\mathcal{T}(\boldsymbol{\epsilon}; \mathbf{v})\right) \nabla_{\mathbf{v}} \left(\log q\left(\mathcal{T}(\boldsymbol{\epsilon}; \mathbf{v}); \mathbf{v}\right) + \log J(\boldsymbol{\epsilon}, \mathbf{v})\right) d\boldsymbol{\epsilon} \tag{10}$$

$$= \int q\left(\mathcal{T}(\boldsymbol{\epsilon}; \mathbf{v}); \mathbf{v}\right) J(\boldsymbol{\epsilon}, \mathbf{v}) f\left(\mathcal{T}(\boldsymbol{\epsilon}; \mathbf{v})\right) \left(\nabla_{\mathbf{v}} \log q\left(\mathcal{T}(\boldsymbol{\epsilon}; \mathbf{v}); \mathbf{v}\right) + \nabla_{\mathbf{v}} \log J(\boldsymbol{\epsilon}, \mathbf{v})\right) d\boldsymbol{\epsilon}. \tag{11}$$

The derivative $\nabla_{\mathbf{v}} \log q \left( \mathcal{T}(\boldsymbol{\epsilon}; \mathbf{v}); \mathbf{v} \right)$ can be obtained by the chain rule. If $\mathbf{z} = \mathcal{T}(\boldsymbol{\epsilon}; \mathbf{v})$, then $\nabla_{\mathbf{v}} \log q \left( \mathcal{T}(\boldsymbol{\epsilon}; \mathbf{v}); \mathbf{v} \right) = \nabla_{\mathbf{z}} \log q(\mathbf{z}; \mathbf{v}) \nabla_{\mathbf{v}} \mathcal{T}(\boldsymbol{\epsilon}; \mathbf{v}) + \nabla_{\mathbf{v}} \log q(\mathbf{z}; \mathbf{v})$. We substitute this result in the above equation and revert the change of variables back to the original space $\mathbf{z}$ (also multiplying by the inverse Jacobian), yielding

$$
\begin{aligned}
\mathbf{g}^{\text{corr}} &= \int q(\mathbf{z}; \mathbf{v}) f(\mathbf{z}) \left( \nabla_{\mathbf{z}} \log q(\mathbf{z}; \mathbf{v}) h \left( \mathcal{T}^{-1}(\mathbf{z}; \mathbf{v}); \mathbf{v} \right) + \nabla_{\mathbf{v}} \log q(\mathbf{z}; \mathbf{v}) + u \left( \mathcal{T}^{-1}(\mathbf{z}; \mathbf{v}); \mathbf{v} \right) \right) d\mathbf{z} \\
&= \mathbb{E}_{q(\mathbf{z}; \mathbf{v})} \left[ f(\mathbf{z}) \left( \nabla_{\mathbf{z}} \log q(\mathbf{z}; \mathbf{v}) h \left( \mathcal{T}^{-1}(\mathbf{z}; \mathbf{v}); \mathbf{v} \right) + \nabla_{\mathbf{v}} \log q(\mathbf{z}; \mathbf{v}) + u \left( \mathcal{T}^{-1}(\mathbf{z}; \mathbf{v}); \mathbf{v} \right) \right) \right],
\end{aligned}
\tag{12}
$$

where we have used the definition of the functions $h(\boldsymbol{\epsilon}; \mathbf{v})$ and $u(\boldsymbol{\epsilon}; \mathbf{v})$.

## 2 Particularization for the Gamma Distribution

For the gamma distribution we choose the transformation

$$
z = \mathcal{T}(\epsilon; \alpha, \beta) = \exp(\epsilon \sqrt{\psi_1(\alpha)} + \psi(\alpha) - \log(\beta)).
\tag{13}
$$

Thus, we have that

$$
J(\epsilon, \alpha, \beta) = |\det \nabla_\epsilon \mathcal{T}(\epsilon; \alpha, \beta)| = \mathcal{T}(\epsilon; \alpha, \beta) \sqrt{\psi_1(\alpha)}.
\tag{14}
$$

The derivatives of $\log q(z; \alpha, \beta)$ with respect to its arguments are given by

$$
\frac{\partial}{\partial z} \log q(z; \alpha, \beta) = \frac{\alpha - 1}{z} - \beta,
\tag{15}
$$

$$
\frac{\partial}{\partial \alpha} \log q(z; \alpha, \beta) = \log(\beta) - \psi(\alpha) + \log(z),
\tag{16}
$$

$$
\frac{\partial}{\partial \beta} \log q(z; \alpha, \beta) = \frac{\alpha}{\beta} - z.
\tag{17}
$$

Therefore, the auxiliary functions $h(\epsilon; \alpha, \beta)$ and $u(\epsilon; \alpha, \beta)$ for the components of the gradient with respect to $\alpha$ and $\beta$ can be written as

$$
h_\alpha(\epsilon; \alpha, \beta) = \frac{\partial}{\partial \alpha} \mathcal{T}(\epsilon; \alpha, \beta) = \mathcal{T}(\epsilon; \alpha, \beta) \left( \frac{\epsilon \psi_2(\alpha)}{2\sqrt{\psi_1(\alpha)}} + \psi_1(\alpha) \right),
\tag{18}
$$

$$
h_\beta(\epsilon; \alpha, \beta) = \frac{\partial}{\partial \beta} \mathcal{T}(\epsilon; \alpha, \beta) = -\frac{\mathcal{T}(\epsilon; \alpha, \beta)}{\beta},
\tag{19}
$$

$$
u_\alpha(\epsilon; \alpha, \beta) = \frac{\partial}{\partial \alpha} \log J(\epsilon, \alpha, \beta) = \left( \frac{\epsilon \psi_2(\alpha)}{2\sqrt{\psi_1(\alpha)}} + \psi_1(\alpha) \right) + \frac{\psi_2(\alpha)}{2\psi_1(\alpha)},
\tag{20}
$$

$$
u_\beta(\epsilon; \alpha, \beta) = \frac{\partial}{\partial \beta} \log J(\epsilon, \alpha, \beta) = -\frac{1}{\beta}.
\tag{21}
$$

Thus, we finally obtain that the components of $\mathbf{g}^{\text{rep}}$ corresponding to the derivatives with respect to $\alpha$ and $\beta$ are given by

$$
\mathbf{g}_\alpha^{\text{rep}} = \mathbb{E}_{q(z; \alpha, \beta)} \left[ \frac{\partial}{\partial z} f(\mathbf{z}) \times z \left( \frac{\mathcal{T}^{-1}(z; \alpha, \beta) \psi_2(\alpha)}{2\sqrt{\psi_1(\alpha)}} + \psi_1(\alpha) \right) \right],
\tag{22}
$$

$$
\mathbf{g}_\beta^{\text{rep}} = \mathbb{E}_{q(z; \alpha, \beta)} \left[ \frac{\partial}{\partial z} f(\mathbf{z}) \times \frac{-z}{\beta} \right],
\tag{23}
$$

while the components of $\mathbf{g}^{\text{corr}}$ can be similarly obtained by substituting the expressions above into Eq. 12. Remarkably, we obtain that

$$
\mathbf{g}_\beta^{\text{corr}} = 0.
\tag{24}
$$

# 3 Particularization for the Beta Distribution

For a random variable $z \sim \text{Beta}(\alpha, \beta)$, we could rewrite $z = z_1'/(z_1' + z_2')$ for $z_1' \sim \text{Gamma}(\alpha, 1)$ and $z_2' \sim \text{Gamma}(\beta, 1)$, and apply the above method for the gamma-distributed variables $z_1'$ and $z_2'$. Instead, in the spirit of applying standardization directly over $z$, we define a transformation to standardize the logit function. This leads to

$$z = \mathcal{T}(\epsilon; \alpha, \beta) = \frac{1}{1 + \exp(-\epsilon\sigma - \psi(\alpha) + \psi(\beta))}. \tag{25}$$

This transformation ensures that $\epsilon$ has mean zero. However, in this case we do not specify the form of $\sigma$, and we let it be a function of $\alpha$ and $\beta$. This allows us to choose $\sigma$ in such a way that $\mathbf{g}^{\text{corr}} = \mathbf{0}$ for the sampled value of $z$, which we found to work well (even though this introduces some bias). For simplicity, we write $\sigma = \exp(\phi)$.

Thus, we have that

$$J(\epsilon, \alpha, \beta) = |\det \nabla_\epsilon \mathcal{T}(\epsilon; \alpha, \beta)| = \mathcal{T}(\epsilon; \alpha, \beta)(1 - \mathcal{T}(\epsilon; \alpha, \beta))\sigma. \tag{26}$$

The derivatives of $\log q(z; \alpha, \beta)$ with respect to its arguments are given by

$$\frac{\partial}{\partial z} \log q(z; \alpha, \beta) = \frac{\alpha - 1}{z} - \frac{\beta - 1}{1 - z}, \tag{27}$$

$$\frac{\partial}{\partial \alpha} \log q(z; \alpha, \beta) = \psi(\alpha + \beta) - \psi(\alpha) + \log(z), \tag{28}$$

$$\frac{\partial}{\partial \beta} \log q(z; \alpha, \beta) = \psi(\alpha + \beta) - \psi(\beta) + \log(1 - z). \tag{29}$$

Therefore, the auxiliary functions $h(\epsilon; \alpha, \beta)$ and $u(\epsilon; \alpha, \beta)$ for the components of the gradient with respect to $\alpha$ and $\beta$ can be written as

$$h_\alpha(\epsilon; \alpha, \beta) = \frac{\partial}{\partial \alpha}\mathcal{T}(\epsilon; \alpha, \beta) = \mathcal{T}(\epsilon; \alpha, \beta)(1 - \mathcal{T}(\epsilon; \alpha, \beta))\left(\psi_1(\alpha) + \epsilon\sigma\frac{\partial\phi}{\partial\alpha}\right), \tag{30}$$

$$h_\beta(\epsilon; \alpha, \beta) = \frac{\partial}{\partial \beta}\mathcal{T}(\epsilon; \alpha, \beta) = \mathcal{T}(\epsilon; \alpha, \beta)(1 - \mathcal{T}(\epsilon; \alpha, \beta))\left(-\psi_1(\beta) + \epsilon\sigma\frac{\partial\phi}{\partial\beta}\right), \tag{31}$$

$$u_\alpha(\epsilon; \alpha, \beta) = \frac{\partial}{\partial \alpha}\log J(\epsilon, \alpha, \beta) = (1 - 2\mathcal{T}(\epsilon; \alpha, \beta))\left(\psi_1(\alpha) + \epsilon\sigma\frac{\partial\phi}{\partial\alpha}\right) + \frac{\partial\phi}{\partial\alpha}, \tag{32}$$

$$u_\beta(\epsilon; \alpha, \beta) = \frac{\partial}{\partial \beta}\log J(\epsilon, \alpha, \beta) = (1 - 2\mathcal{T}(\epsilon; \alpha, \beta))\left(-\psi_1(\beta) + \epsilon\sigma\frac{\partial\phi}{\partial\beta}\right) + \frac{\partial\phi}{\partial\beta}. \tag{33}$$

Note that the term $\epsilon\sigma$ above can be computed from $z$ without knowledge of the value of $\sigma$ as $\epsilon\sigma = \mathcal{T}^{-1}(z; \alpha, \beta)\sigma = \frac{\text{logit}(z) - \psi(\alpha) + \psi(\beta)}{\sigma}\sigma = \text{logit}(z) - \psi(\alpha) + \psi(\beta)$.

Thus, we finally obtain that the components of $\mathbf{g}^{\text{rep}}$ corresponding to the derivatives with respect to $\alpha$ and $\beta$ are given by

$$\mathbf{g}_\alpha^{\text{rep}} = \mathbb{E}_{q(z;\alpha,\beta)}\left[\frac{\partial}{\partial z}f(\mathbf{z}) \times z(1 - z)\left(\psi_1(\alpha) + (\text{logit}(z) - \psi(\alpha) + \psi(\beta))\frac{\partial\phi}{\partial\alpha}\right)\right], \tag{34}$$

$$\mathbf{g}_\beta^{\text{rep}} = \mathbb{E}_{q(z;\alpha,\beta)}\left[\frac{\partial}{\partial z}f(\mathbf{z}) \times z(1 - z)\left(-\psi_1(\beta) + (\text{logit}(z) - \psi(\alpha) + \psi(\beta))\frac{\partial\phi}{\partial\beta}\right)\right], \tag{35}$$

where we are still free to choose $\partial\phi/\partial\alpha$ and $\partial\phi/\partial\beta$. We have found that the choice of these values such that $\mathbf{g}_\alpha^{\text{corr}} = \mathbf{g}_\beta^{\text{corr}} = 0$ works well in practice. Thus, we set the derivatives of $\phi$ such that the relationships

$$\frac{\partial}{\partial z} \log q(z; \alpha, \beta) \times h_\alpha\left(\mathcal{T}^{-1}(z; \alpha, \beta); \alpha, \beta\right) + \frac{\partial}{\partial \alpha} \log q(z; \alpha, \beta) + u_\alpha\left(\mathcal{T}^{-1}(z; \alpha, \beta); \alpha, \beta\right) = 0, \tag{36}$$

$$\frac{\partial}{\partial z} \log q(z; \alpha, \beta) \times h_\beta\left(\mathcal{T}^{-1}(z; \alpha, \beta); \alpha, \beta\right) + \frac{\partial}{\partial \beta} \log q(z; \alpha, \beta) + u_\beta\left(\mathcal{T}^{-1}(z; \alpha, \beta); \alpha, \beta\right) = 0, \tag{37}$$

hold for the sampled value of $z$. This involves solving a simple linear equation for $\partial\phi/\partial\alpha$ and $\partial\phi/\partial\beta$.

# 4 Particularization for the Dirichlet Distribution

For a Dirichlet($\boldsymbol{\alpha}$) distribution, with $\boldsymbol{\alpha} = [\alpha_1, \ldots, \alpha_K]$, we can apply the standardization

$$\mathbf{z} = \mathcal{T}(\boldsymbol{\epsilon}; \boldsymbol{\alpha}) = \exp\left(\boldsymbol{\Sigma}^{1/2}\boldsymbol{\epsilon} + \boldsymbol{\mu}\right), \tag{38}$$

where the mean $\boldsymbol{\mu}$ is a $K$-length vector and the covariance $\boldsymbol{\Sigma}$ is a $K \times K$ matrix,[1,2] which are respectively given by

$$\boldsymbol{\mu} = \mathbb{E}_{q(\mathbf{z};\boldsymbol{\alpha})}[\log(\mathbf{z})] = \begin{bmatrix} \psi(\alpha_1) - \psi(\alpha_0) \\ \vdots \\ \psi(\alpha_K) - \psi(\alpha_0) \end{bmatrix} \tag{39}$$

and

$$(\boldsymbol{\Sigma})_{ij} = \mathrm{Cov}(\log(z_i), \log(z_j)) = \begin{cases} \psi_1(\alpha_i) - \psi_1(\alpha_0) & \text{if } i = j, \\ -\psi_1(\alpha_0) & \text{if } i \neq j. \end{cases} \tag{40}$$

Here, we have defined $\alpha_0 = \sum_k \alpha_k$. The covariance matrix $\boldsymbol{\Sigma}$ can be rewritten as a diagonal matrix plus a rank one update, which can be exploited for faster computations:

$$\boldsymbol{\Sigma} = \mathrm{diag}\left(\begin{bmatrix} \psi_1(\alpha_1) \\ \vdots \\ \psi_1(\alpha_K) \end{bmatrix}\right) - \psi_1(\alpha_0)\mathbf{1}\mathbf{1}^\top. \tag{41}$$

Note that, since $\boldsymbol{\Sigma}$ is positive semidefinite, $\boldsymbol{\Sigma}^{1/2}$ can be readily obtained after diagonalization. In other words, if we express $\boldsymbol{\Sigma} = \mathbf{V}\mathbf{D}\mathbf{V}^\top$, where $\mathbf{V}$ is an orthonormal matrix and $\mathbf{D}$ is a diagonal matrix, then $\boldsymbol{\Sigma}^{1/2} = \mathbf{V}\mathbf{D}^{1/2}\mathbf{V}^\top$.

Given the transformation above, we can write

$$J(\boldsymbol{\epsilon}, \boldsymbol{\alpha}) = |\det \nabla_{\boldsymbol{\epsilon}} \mathcal{T}(\boldsymbol{\epsilon}; \boldsymbol{\alpha})| = \det(\boldsymbol{\Sigma}^{1/2}) \prod_i \mathcal{T}_i(\boldsymbol{\epsilon}; \boldsymbol{\alpha}). \tag{42}$$

The derivatives of $\log q(\mathbf{z}; \boldsymbol{\alpha})$ with respect to its arguments are given by

$$\frac{\partial}{\partial z_i} \log q(\mathbf{z}; \boldsymbol{\alpha}) = \frac{\alpha_i - 1}{z_i}, \tag{43}$$

$$\frac{\partial}{\partial \alpha_i} \log q(\mathbf{z}; \boldsymbol{\alpha}) = \psi(\alpha_0) - \psi(\alpha_i) + \log(z_i). \tag{44}$$

Therefore, the auxiliary functions $h(\boldsymbol{\epsilon}; \boldsymbol{\alpha})$ and $u(\boldsymbol{\epsilon}; \boldsymbol{\alpha})$ can be written as

$$h(\boldsymbol{\epsilon}; \boldsymbol{\alpha}) = \nabla_{\boldsymbol{\alpha}} \mathcal{T}(\boldsymbol{\epsilon}; \boldsymbol{\alpha}) = \begin{bmatrix} \mathcal{T}_1(\boldsymbol{\epsilon}; \boldsymbol{\alpha})\left(\frac{\partial(\boldsymbol{\Sigma}_{1:}^{1/2})}{\partial \alpha_1}\boldsymbol{\epsilon} + \frac{\partial \mu_1}{\partial \alpha_1}\right) & \cdots & \mathcal{T}_1(\boldsymbol{\epsilon}; \boldsymbol{\alpha})\left(\frac{\partial(\boldsymbol{\Sigma}_{1:}^{1/2})}{\partial \alpha_K}\boldsymbol{\epsilon} + \frac{\partial \mu_1}{\partial \alpha_K}\right) \\ \vdots & \ddots & \vdots \\ \mathcal{T}_K(\boldsymbol{\epsilon}; \boldsymbol{\alpha})\left(\frac{\partial(\boldsymbol{\Sigma}_{K:}^{1/2})}{\partial \alpha_1}\boldsymbol{\epsilon} + \frac{\partial \mu_K}{\partial \alpha_1}\right) & \cdots & \mathcal{T}_K(\boldsymbol{\epsilon}; \boldsymbol{\alpha})\left(\frac{\partial(\boldsymbol{\Sigma}_{K:}^{1/2})}{\partial \alpha_K}\boldsymbol{\epsilon} + \frac{\partial \mu_K}{\partial \alpha_K}\right) \end{bmatrix}, \tag{45}$$

$$u(\boldsymbol{\epsilon}; \boldsymbol{\alpha}) = \nabla_{\boldsymbol{\alpha}} \log J(\boldsymbol{\epsilon}, \boldsymbol{\alpha}) = \begin{bmatrix} \frac{\partial \log \det(\boldsymbol{\Sigma}^{1/2})}{\partial \alpha_1} + \sum_i \left(\frac{\partial(\boldsymbol{\Sigma}_{i:}^{1/2})}{\partial \alpha_1}\boldsymbol{\epsilon} + \frac{\partial \mu_i}{\partial \alpha_1}\right) \\ \vdots \\ \frac{\partial \log \det(\boldsymbol{\Sigma}^{1/2})}{\partial \alpha_K} + \sum_i \left(\frac{\partial(\boldsymbol{\Sigma}_{i:}^{1/2})}{\partial \alpha_K}\boldsymbol{\epsilon} + \frac{\partial \mu_i}{\partial \alpha_K}\right) \end{bmatrix}. \tag{46}$$

The intermediate derivatives that are necessary for the computation of the functions $h(\epsilon; \boldsymbol{\alpha})$ and $u(\epsilon; \boldsymbol{\alpha})$ are:

$$\frac{\partial \boldsymbol{\mu}}{\partial \alpha_i} = \begin{bmatrix} -\psi_1(\alpha_0) \\ -\psi_1(\alpha_0) \\ \vdots \\ \psi_1(\alpha_i) - \psi_1(\alpha_0) \\ \vdots \\ -\psi_1(\alpha_0) \end{bmatrix} \tag{47}$$

$$\frac{\partial \log \det(\boldsymbol{\Sigma}^{1/2})}{\partial \alpha_i} = \text{trace}\left(\boldsymbol{\Sigma}^{-1/2} \frac{\partial \boldsymbol{\Sigma}^{1/2}}{\partial \alpha_i}\right), \tag{48}$$

and $\frac{\partial \boldsymbol{\Sigma}^{1/2}}{\partial \alpha_i}$ is the solution to the Lyapunov equation

$$\frac{\partial \boldsymbol{\Sigma}}{\partial \alpha_i} = \frac{\partial \boldsymbol{\Sigma}^{1/2}}{\partial \alpha_i} \boldsymbol{\Sigma}^{1/2} + \boldsymbol{\Sigma}^{1/2} \frac{\partial \boldsymbol{\Sigma}^{1/2}}{\partial \alpha_i}, \tag{49}$$

where

$$\frac{\partial \boldsymbol{\Sigma}}{\partial \alpha_i} = \text{diag}\left(\begin{bmatrix} 0 \\ 0 \\ \vdots \\ \psi_2(\alpha_i) \\ \vdots \\ 0 \end{bmatrix}\right) - \psi_2(\alpha_0)\mathbf{1}\mathbf{1}^\top. \tag{50}$$

Putting all this together, we finally have the expressions for the generalized reparameterization gradient:

$$\mathbf{g}^{\text{rep}} = \mathbb{E}_{q(\mathbf{z};\boldsymbol{\alpha})}\left[h^\top\left(\mathcal{T}^{-1}(\mathbf{z};\boldsymbol{\alpha});\boldsymbol{\alpha}\right) \nabla_{\mathbf{z}} f(\mathbf{z})\right], \tag{51}$$

$$\mathbf{g}^{\text{corr}} = \mathbb{E}_{q(\mathbf{z};\boldsymbol{\alpha})}\left[f(\mathbf{z})\left(h^\top\left(\mathcal{T}^{-1}(\mathbf{z};\boldsymbol{\alpha});\boldsymbol{\alpha}\right) \nabla_{\mathbf{z}} \log q(\mathbf{z};\mathbf{v}) + \nabla_{\boldsymbol{\alpha}} \log q(\mathbf{z};\boldsymbol{\alpha}) + u\left(\mathcal{T}^{-1}(\mathbf{z};\boldsymbol{\alpha});\boldsymbol{\alpha}\right)\right)\right], \tag{52}$$

## 5  Experimental Results

### 5.1  Using more than 1 sample

We now study the sensitivity of the generalized reparameterization gradient with respect to the number of samples of the Monte Carlo estimator. For that, we choose the Olivetti dataset, and we apply the generalized reparameterization approach using 2, 5, 10, and 20 Monte Carlo samples. At each iteration, we compute the evidence lower bound (ELBO) and the average sample variance of the gradient estimator. We report these results in Figure 1 for the first 200 iterations of the inference procedure. As expected, increasing the number of samples is beneficial because it reduces the resulting variance. The gap between the curves with 10 and 20 samples is negligible, specially after 100 iterations. A larger number of samples seems to be particularly helpful in the very early iterations of inference.

### 5.2  Reconstructed images

Here, we show some reconstructed observations for the three datasets involving images, namely, the binarized MNIST, the Olivetti dataset, and Omniglot. We plot the reconstructed images as follows: we first draw one sample from the variational posterior, and then we compute the *mean* of the observations for that particular sample of latent variables.

Figure 2 shows the results for the Olivetti dataset. The true observations are shown in the left panel, whereas the corresponding reconstructed images are shown in the center panel (for G-REP) and the right panel (for ADVI). We can observe that the images obtained from G-REP are more detailed (e.g., we can distinguish the glasses, mustache, or facial expressions) than the images obtained from ADVI.

**(a)** ELBO.          **(b)** Average variance.

**Figure 1:** Performance of G-REP for different number of Monte Carlo samples.

We argue that this effect is due to the variational family used by automatic differentiation variational inference (ADVI), which cannot capture well sparse posterior distributions, for which samples close to 0 are common.

This behavior is similar in the case of the digits from MNIST or the characters from Omniglot. We show these images in Figures 3 and 4, respectively. Once again, images sampled from the G-REP posterior are visually closer to the ground truth that images sampled from the ADVI posterior, which tend to be more blurry, or even unrecognizable in a few cases.

**(a)** True observations.          **(b)** Reconstructed (G-REP).          **(c)** Reconstructed (ADVI).

**Figure 2:** Images from the Olivetti dataset. ADVI provides less detailed images when compared to G-REP.

(a) True observations.  (b) Reconstructed (G-REP).  (c) Reconstructed (ADVI).

**Figure 3:** Images from the binarized MNIST dataset. ADVI provides more blurry images when compared to G-REP.

(a) True observations.  (b) Reconstructed (G-REP).  (c) Reconstructed (ADVI).

**Figure 4:** Images from the Omniglot dataset. ADVI provides more blurry images when compared to G-REP.

## Footnotes

[1] Instead, we could define a transformation that ignores the off-diagonal terms of the covariance matrix. This would lead to faster computations but higher variance of the resulting estimator.

[2] We could also apply the full-covariance transformation for the beta distribution.