[Reviews · NeurIPS 2016]

Reviewer 1

Summary

This works studies stochastic variational inference for approximate posterior inference. The authors propose a unified framework for black-box variational inference based on the log-derivative trick (or score function method) and the reparametrization trick. More specifically, they generalise the latter to non-Gaussian posteriors and discuss three other popular cases (Gamma, Beta and log-normal). The report good experimental results.

Qualitative Assessment

This paper was a pleasure to read. It was very well-written and provided an excellent and comprehensive synthesis of recent advances that make stochastic variational inference more generic. The proposed formulation unifies black-box variational inference based on the score function and the reparametrization trick (used in the variational auto-encoder). The authors introduce the so-called generalized reparameterization gradient, a method that extends the reparameterization gradient to a wider class of variational distributions by allowing for weak dependency. The main idea is to define an invertible transformation of the latent variables such that the distribution of the transformed variables is only weakly governed by the variational parameters. This essentially means that at least the first moments of the auxiliary variable do not depend on the original variational parameters. While being presented as general, it is still required in practice to identify an appropriate transformation. Also, the considered cases are in the same class as the one covered by ADVI, but results are encouraging. The general reformulation is also a step forward in the direction of handling non-conjugate cases. Could the authors discuss the choice of the denominator in the case of the beta into more detail? From the appendix, it was not clear to me why these choices are working well. While the authors indicate that it is sufficient to consider only one sample, could they comment on how the results improve as a function of the number of samples?

Confidence in this Review

3-Expert (read the paper in detail, know the area, quite certain of my opinion)


Reviewer 2

Summary

The paper proposes a generalization of the reparameterization trick beyond the Gaussian distribution which is used in variational inference such as VAE. This is achieved by defining an invertible transformation of the latent variables for which the variational parameters are weakly dependent. In the experiments, the proposed method is demonstrated in two nonconjugate probabilistic models involving gamma and beta latent variables.

Qualitative Assessment

I really enjoyed the paper and the idea. The paper is very well and clearly written. The proposed method seems to be useful and broaden the usage of the recent advances in variational inference.

Confidence in this Review

2-Confident (read it all; understood it all reasonably well)


Reviewer 3

Summary

The author extended reparameterization gradients, a technique of taking gradient for variational inference that assumes the Gaussianity of the variational distribution, to more general framework that allows non-Gaussian variational distributions.

Qualitative Assessment

The past literature, motivation, and contributions are clearly described in Introduction. Major comments: 1. Variance reduction is not verified. In lines 154--162 you explain the motivation of introducing the form (9) that aims to reduce the variance of the estimator of the gradient so that the speed of convergence becomes faster. You also explain that when g^corr gets small the variance may get small as well. However, I cannot understand the logic here: why the variance should be small when g^corr is small? Some theoretical analysis or at least high-level explanation are necessary. Furthermore, the reduction of the variance was not confirmed even in an empirical way. The authors should check this by e.g. using synthetic data. 2. The range of application is not fully explained. This method requires several assumptions, such as z is continuous and f is differentiable w.r.t. z. The class of q seems to be restricted as well. However, the clear summarization of them is not presented in this manuscript. The authors should summarize them clearly so that readers can easily check whether this approach is applicable to their problem. Minor comments: A. The recovered images reported in Supplementary material are helpful to assess the quality of inference. It is better to show the results of BBVI as well as G-REP and ADVI. B. If q is exponential family, can we gain additional benefits (e.g. analytic solution of something) in your approach? C. In the experiments, as a baseline method, it would be better to compare with standard variational inference with Gaussian q.

Confidence in this Review

2-Confident (read it all; understood it all reasonably well)


Reviewer 4

Summary

This paper proposes a so-called generalization of the reparameterization trick. In fact, the gradient has been described before, and the paper seriously misrepresents the state of the art and the contributions made.

Qualitative Assessment

Second line of abstract: "[the reparameterization gradient] only applies when fitting approximate Gaussian distributions". This is a false statement: as is explained in the "auto-encoding variational bayes" (AEVB) paper, reparameterization applies to a much broader set of families: location-scale families, distribution with tractable inverse CDF, etc. Please see that paper. One of the three examples given in the paper, namely the log-normal distribution, can be optimized using the standard reparameterization trick. Furthermore, the gradient estimator has been described fully by Schulman et al (2015) for general stochastic computation graphs. The paper does some good experimental results for VI and discusses the Beta and Gamma distributions, however these contributions are too limited for publication. The paper is clearly written.

Confidence in this Review

3-Expert (read the paper in detail, know the area, quite certain of my opinion)


Reviewer 5

Summary

This paper proposed the generalization of the reparameterization trick proposed by Kingma and Welling. They derived reparameterization gradients for several distributions. They provided empirical results of their algorithms.

Qualitative Assessment

The generalization of a reparameterization trick is important and interesting in NIPS communities. However, the definition of each transformation seems to be specific to each distribution. Could you provide the general derivation of the transformation of exponential families.

Confidence in this Review

2-Confident (read it all; understood it all reasonably well)


Reviewer 6

Summary

This paper generalizes the reparameterization trick for variational inference to a broader distribution family. The key idea is to decompose the gradient of expected objective function into two parts: one represents the standard reparameterization gradient, the other one is a correction term due to the induced dependency of transformation. The main contribution I assumed should be how to make the correction term small.

Qualitative Assessment

This paper is well written and easy to follow, except for one missing definition, \Psi_k: polygamma function. In Kingma's original VAE paper, it summarizes three tricks for most distributions: tractable inverse CDF, reparameterization trick for location-scale family, and composition. This paper makes it possible for some distributions that used to require composition trick, such as gamma distribution. The contribution of this paper seems incremental, but the technique is important and will definitely benefit the variational inference community. The detail in constructing adaptive step-size SGD (sec 3.2) can be put in supplementary materials, while the author can emphasize more on the correction term. In addition, the author can list other possible distributions that this trick can apply in supplements. The running time in experiments is acceptable, but I'd like to see the scalability of this method, e.g. the author should provide the result on larger dataset, at least the full MNIST.

Confidence in this Review

3-Expert (read the paper in detail, know the area, quite certain of my opinion)